# ACE: Attack Combo Enhancement Against Machine Learning Models

## Abstract

Machine learning (ML) models are proving to be vulnerable to a variety of attacks that allow the adversary to learn sensitive information, cause mispredictions, and more. While these attacks have been extensively studied, current research predominantly focuses on analyzing each attack type individually. In practice, however, adversaries may employ multiple attack strategies simultaneously rather than relying on a single approach. This prompts a crucial yet underexplored question: when the adversary has multiple attacks at their disposal, are they able to mount or enhance the effect of one attack with another? In this paper, we take the first step in studying the *intentional interactions* among different attacks, which we define as attack combos. Specifically, we focus on four well-studied attacks during the model's inference phase: adversarial examples, attribute inference, membership inference, and property inference. To facilitate the study of their interactions, we propose a taxonomy based on three stages of the attack pipeline: preparation, execution, and evaluation. Using this taxonomy, we identify four effective attack combos, such as property inference assisting attribute inference at its preparation level and adversarial examples assisting property inference at its execution level. We conduct extensive experiments on the attack combos using three ML model architectures and three benchmark image datasets. Empirical results demonstrate the effectiveness of these four attack combos. We implement and release a modular reusable toolkit, ACE. Arguably, our work serves as a call for researchers and practitioners to consider advanced adversarial settings involving multiple attack strategies, aiming to strengthen the security and robustness of AI systems.

## 1 Introduction

Recently, machine learning has gained momentum in multiple fields, achieving success in real-world deployments, such as image classification (Devlin et al., 2019; Bao et al., 2020; Zhang et al., 2021), face recognition (Zheng et al., 2017; Kemelmacher-Shlizerman et al., 2016), and medical image analysis (Kourou et al., 2015; Stanfill et al., 2010; Burlina et al., 2011). Nevertheless, prior research has shed light on the vulnerability of ML models to various attacks, such as adversarial examples (Iyyer et al., 2018; Ribeiro et al., 2018; Alzantot et al., 2018), membership inference (Shokri et al., 2017; Nasr et al., 2018; Salem et al., 2019; Li & Zhang, 2021), and backdoor attacks (Chen et al., 2017; Gu et al., 2017; Liu et al., 2018). These vulnerabilities prompt significant security and privacy risks. As a result, investigating, quantifying, and mitigating these various attacks on ML models have become increasingly important topics.

Currently, most research in this field focuses on developing or optimizing more powerful attacks, e.g., higher attack success rates or greater stealthiness, and proposing corresponding countermeasures. More precisely, these studies typically focus on individual attacks. While some measurement or benchmark papers exist that consider multiple attacks, e.g., ML-Doctor (Liu et al., 2022b) or SecurityNet (Zhang et al., 2024), they still implement each attack individually. In other words, studying attacks in isolation is actually the most common practice in the existing ML security domain.

However, this practice may not accurately reflect real-world scenarios, where adversaries often possess multiple attack strategies and can potentially synergize or leverage them simultaneously. When focusing solely on individual attacks, researchers may overlook the potential for adversaries to amplify the impact of one attack by leveraging knowledge or capabilities gained from another attack.

Consequently, the true extent of vulnerabilities and risks posed by combined attacks may be underestimated or remain unexplored.

This reality prompts the need for a more comprehensive understanding of the *intentional interactions* among different attacks.

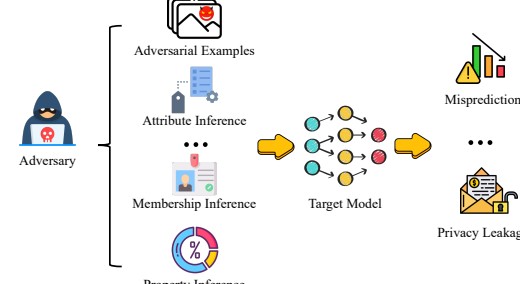

### 1.1 CONTRIBUTIONS

In this work, we take the first step in exploring the (possible) intentional interactions between different types of attacks. We focus exclusively on the inference phase of ML models

**Figure 1:** Given a target model, the adversary can launch different attacks to achieve different malicious goals.

since deployed models are more likely to face intentional interactions between different attacks. Specifically, we consider the four most representative attacks launched during the ML model's inference phase, aka *inference-time attacks*: adversarial examples (Iyyer et al., 2018; Ribeiro et al., 2018; Alzantot et al., 2018), attribute inference (Melis et al., 2019; Song & Shmatikov, 2020), membership inference (Shokri et al., 2017; Nasr et al., 2018; Salem et al., 2019; Li & Zhang, 2021), and property inference (Melis et al., 2019).

We formulate the following research questions (RQs), targeting addressing this significant gap:

- **RQ1:** How can we approach the design and implementation of attack combos?
- **RQ2:** How can the knowledge gained from one type of attack facilitate or enhance the effectiveness of another attack?
- **RQ3:** How effective are combined attacks in exploiting ML model vulnerabilities compared to individual ones?

**Combo Taxonomy.** First, we propose a taxonomy for attack combinations based on the attack pipeline (**RQ1**), divided into three levels: preparation, execution, and evaluation. The former encompasses all preliminary activities before the main attack, including tool setup, data collection, and configuration. The execution level covers the attack's actual implementation, involving malicious queries, responses, and vulnerability exploitation. Finally, the evaluation level assesses the attack impact, including system disruption, goal achievement, and any post-exploitation activities.

**Combo Methodology.** Based on the taxonomy, we conduct an extensive exploration of attack combos across four representative inference-time attacks (**RQ2**). Specifically, we identify four effective attack combos: one at the preparation level, two at the execution level, and one at the assessment level. At the preparation level, we propose using property inference to assist attribute inference (PropInf2AttrInf). By determining the attribute distribution in the victim model's training dataset through property inference, we use it to create a balanced attack training dataset for attribute inference. At the execution level, we propose two attack combos: using adversarial examples to assist membership inference (ADV2MemInf) and property inference (ADV2PropInf), respectively. Adversarial examples can search for different noise magnitudes for various membership or property statuses, which are then integrated into their original information for improved attack performance. At the evaluation level, we leverage property inference to assist membership inference (PropInf2MemInf). After the membership inference process ends, we use the property distribution determined by property inference to calibrate its attack output.

**Combo Evaluation.** We conduct extensive experiments across three popular ML model architectures and three benchmark image datasets (**RQ3**). We here summarize our analysis using ResNet18 (He et al., 2016) trained on CIFAR10 (Krizhevsky, 2009) as an example. First, property inference significantly enhances attribute inference at its preparation level. For instance, AttrInf achieves an accuracy of 0.500 while PropInf2AttrInf achieves an empirical accuracy of 0.894 and a theoretical accuracy of 0.872. Second, adversarial examples improve both membership inference and property inference. For instance, the black-box MemInf with shadow model and PropInf achieve an accuracy of 0.664 and 0.890, respectively, while the attack combos yield significantly improved results, with accuracies of 0.851 and 0.960, respectively. Finally, the black-box MemInf with partial training dataset achieves an accuracy of 0.631, compared to PropInf2MemInf's accuracy of 0.669.

**ACE.** To evaluate our proposed diverse attack combos, we develop a modular framework, ACE (Attack Combo Enhancement). With its modular design, ACE allows for easy integration of new versions of each attack type, additional datasets, and models. Our code will be released publicly along with the final version of the paper (and is already available upon request), thus facilitating further research in the field.

## 2 THREAT MODELING

This work focuses on image classification ML models, where the model takes a data sample as input and outputs a probability vector, known as posteriors. Each component of the posteriors represents the likelihood that the sample belongs to a specific class.

We categorize the threat models along two dimensions: 1) *access to the target model* and 2) *availability of an auxiliary dataset*.

**Access to the Target Model.** We consider two access settings: *white-box* and *black-box*. In the white-box setting ($\mathcal{M}^{\mathsf{W}}$), the adversary has full knowledge of the target model, including its parameters and architecture. In contrast, the black-box setting ($\mathcal{M}^{\mathsf{B}}$) limits the adversary to interact with the model like an API, where they can only query it and receive outputs. However, much of the black-box literature (Shokri et al., 2017; Ganju et al., 2018; Xu et al., 2021) also assumes the adversary knows the model's architecture, which they use to build shadow models (see Appendix A).

**Auxiliary Dataset.** The adversary needs an auxiliary dataset to train their attack model. For this knowledge, we consider three scenarios: 1) *partial training dataset* ($\mathcal{D}^{\mathsf{P}}_{\text{aux}}$), 2) *shadow auxiliary dataset* ($\mathcal{D}^{\mathsf{S}}_{\text{aux}}$), and 3) *query auxiliary dataset* ($\mathcal{D}^{\mathsf{Q}}_{\text{aux}}$). In the first scenario, the adversary acquires part of the real training data of the target model (datasets where it is public knowledge). For the $\mathcal{D}^{\mathsf{S}}_{\text{aux}}$ setting, where the adversary gets a "shadow" dataset from the same distribution as the training data of the target model, which is used to train a shadow model (see Section V-C in (Shokri et al., 2017) for a discussion on how to generate such data). In the last scenario, the adversary establishes a dataset with different property proportions to query the shadow model, thereby training the attack model for PropInf. This dataset is never used to train either the target model or the shadow model, and it needs to have the same distribution as the target training dataset. Unlike the first two settings, $\mathcal{D}^{\mathsf{Q}}_{\text{aux}}$ is constructed based on the second property proportions that may exist during model training (see Appendix A.4).

## 3 ATTACK COMBO

In this section, we introduce our hierarchical combinations of different attack types. First, we propose a taxonomy that offers a structured framework for studying these combinations. Next, we outline the methodologies for specific attack combinations, designating one as the *primary attack* and enhancing it with a *support attack*.

### 3.1 ATTACK COMBO TAXONOMY

To address **RQ1**, which examines the approaches for designing and implementing attack combinations, we propose a taxonomy based on the attack pipeline. This taxonomy serves several purposes: (1) Most attack pipelines consist of multiple phases, allowing integration and combination of different attacks at various phases. (2) It is both domain- and model-agnostic, making it easily adaptable to other areas, such as graph data, NLP, and transformer-based models. (3) It offers future researchers a clear framework for studying attack combinations, providing potential benefits to the community.

**Preparatory Level.** In the preparation stage, the adversary gathers information, sets up the environment, and develops the necessary tools. This includes collecting data about the target machine learning system, such as input-output pairs, model parameters, and any accessible metadata, to understand its architecture. The adversary develops or selects appropriate attack algorithms, like FGSM (Goodfellow et al., 2015) in adversarial example attack, and sets up frameworks and libraries, like PyTorch (https://pytorch.org) or CleverHans (Papernot et al., 2018). Additionally, the adversary prepares the computational infrastructure, including high-performance GPUs or cloud services, and may train a shadow/surrogate model to simulate the target system.

**Execution Level.** During the execution phase, the actual attack is executed against the target machine learning system. For example, the adversary may deploy the attack by generating adversarial examples through perturbing input data to mislead target models or replicating the target model via model extraction. Throughout this phase, the adversary collects outputs and logs detailed data from the target system for subsequent analysis.

**Evaluation Level.** In the evaluation phase, the adversary analyzes the outcomes, assesses the attack performance, and identifies areas for improvement. This involves defining and measuring success metrics such as misclassification rates or confidence reductions, and assessing the broader impact on system performance and security. Post-attack analysis includes examining the types of errors induced by the attack and studying changes in model behavior to understand vulnerabilities. Insights gained during this phase guide the refinement and iteration of the attack strategy, enhancing its effectiveness in subsequent attempts.

## 3.2 Preparation Level

We first introduce attack combinations at the preparatory stage. Here, the support attack supports the primary attack during preparation before the primary attack is executed.

**PropInf2AttrInf**. The first attack combination is enhancing AttrInf (primary attack) by using PropInf (support attack) during its preparatory stage. Specifically, adversaries in AttrInf often overlook a key issue: creating a more effective auxiliary dataset for training attack models. The target attribute bias of the target model's training dataset can complicate the auxiliary dataset, making it crucial to address this bias during preparation. Therefore, we enhance AttrInf by employing PropInf to assist in dataset construction during the preparatory phase.

In general, our intuition is that PropInf can better assist in determining the proportion of the target attribute in the training dataset. For AttrInf, we believe that adversaries will not really care about the proportion of the target attribute in the auxiliary dataset. They can never fully eliminate the influence of the bias in the target model without knowing the property information. Therefore, we first determine the distribution of the target attribute in the training dataset using PropInf and further sample the auxiliary dataset, significantly enhancing the effectiveness of AttrInf. In general, the PropInf2AttrInf can be defined as:

$$\mathsf{PropInf2AttrInf} : x_{\mathsf{target}}, \mathcal{M}^{\mathsf{W}}, \mathcal{D}_{\mathsf{aux}}, \mathsf{PropInf} \to \{\textit{target attributes}\} \tag{1}$$

More concretely, we have two different scenarios for utilizing PropInf, i.e., empirical and theoretical settings. 1) For the empirical setting, we use the real posterior of the PropInf attack model as the confidence for sampling the AttrInf training dataset. For the proportion of the property $p$, given the confidence $c$, the ratio of sampling is $c \times (1 - p)$. 2) On the other hand, for the theoretical setting, we directly use the predicted label from PropInf into the sampling function. In general, when enough shadow models are trained, such as 1,000 for each label, the empirical setting becomes the theoretical setting.

## 3.3 Execution Level

At the execution level, the support attack interacts simultaneously with the primary attack during its execution. This concurrent interaction can amplify the impact of the primary attack by leveraging the synergistic effects of support attacks.

**ADV2MemInf**. Previous work (Li & Zhang, 2021) has demonstrated a distribution shift between the members and non-members when calculating the distance between the adversarial examples and the original images. Following this intuition, we trade this distance as additional information to assist MemInf. For the $\langle \mathsf{MemInf}, \mathcal{M}^{\mathsf{B}}, \mathcal{D}_{\mathsf{aux}} \rangle$, we choose a black-box adversarial attacks, Square (Andriushchenko et al., 2020). Square is a score-based black-box adversarial attack that does not rely on a local gradient. Instead, it utilizes a randomized search scheme that selects localized square-shaped updates at random positions so that at each iteration, the perturbation is situated approximately at the boundary of the dataset. For the $\langle \mathsf{MemInf}, \mathcal{M}^{\mathsf{W}}, \mathcal{D}_{\mathsf{aux}} \rangle$, we choose a white-box adversarial attack, PGD (Madry et al., 2018). It is an iterative method that makes small modifications to the input data at each step by computing the gradient of the loss function with respect to the input data. This gradient demonstrates how to change the input slightly to increase the loss. When the noise $\delta$ added

by the Square or PGD is able to change the prediction of the original label, we stop adding noise and use the data $x_{\text{adv}} = x_{\text{target}} + \delta$ as adversarial examples.

Therefore, we first calculate the $L_2$ distance between member (non-member) samples and their adversarial samples in the auxiliary dataset $\mathcal{D}_{\text{aux}}$. Next, in addition to the normal inputs required for MemInf, such as outputs from the target or shadow model and predicted labels, we also use the $L_2$ distances as other inputs to train the attack model. As a result, ADV2MemInf can be defined as:

$$\text{ADV2MemInf} : x_{\text{target}}, \mathcal{M}, \mathcal{D}_{\text{aux}}, L_2^{\mathcal{D}_{\text{aux}}} \rightarrow \{member, non\text{-}member\} \tag{2}$$

**ADV2PropInf**. Currently, PropInf heavily depends on training a large number of shadow models. The more shadow models, the better the effectiveness of PropInf. However, training such a large number of shadow models is computationally expensive. Therefore, we hope to find additional information to reduce the number of shadow models and increase the accuracy of PropInf. Thus, similar to ADV2MemInf, our intuition is, for the auxiliary datasets $\mathcal{D}_{\text{aux}}^{\text{T}}$ with different proportions of the target property, the distribution of the $L_2$ distance between these samples and their adversarial samples should also be different. For example, the distributions of $L_2$ distances calculated on the auxiliary dataset by models trained on a male-to-female ratio of 5:5 versus 2:8 are different. Following this intuition, we concatenate these $L_2$ distances with the original inputs of PropInf together to train a meta-classifier. ADV2PropInf can be defined as:

$$\text{ADV2PropInf} : \mathcal{M}, \mathcal{D}_{\text{aux}}^{\text{Q}}, \mathcal{D}_{\text{aux}}^{\text{S}}, L_2^{\mathcal{D}_{\text{aux}}^{\text{Q}}} \rightarrow \{target\ property\} \tag{3}$$

### 3.4 EVALUATION LEVEL

At the evaluation stage, the support attack aids the primary attack after its initial execution. This post-attack support can refine the primary attack's outcomes, correct discrepancies, or further exploit vulnerabilities. In other words, the support attack serves to *calibrate* the results of the primary attack.

**PropInf2MemInf**. Previous work (Zhou et al., 2022) finds that PropInf on GAN models can improve the effectiveness of MemInf. MemInf is enhanced by calibrating the output of the attack model with the proportion of the target property $\lambda_p \frac{1}{N} \sum_i^N (\mathcal{P}_i - 0.5)$. Among that, $\lambda_p$ controls the magnitude of the enhancement. $\mathcal{P}_i - 0.5$ is the proportion of the label to which the target sample belongs. However, for the ML models, this calibration is equivalent to directly finding another threshold to classify MemInf. In this scenario, our intuition is a sample has a larger possibility of being a member when it shares the same property with most samples in the target property. Unlike previous work (Zhou et al., 2022), we further train an encoder $\mathcal{E}$ to select different $\lambda$s for the calibration during the attack model training phase, thereby boosting MemInf more effectively. Note that the input of the encoder is the output of the target model $\mathcal{M}$. Formally, the new calibration of MemInf is defined as:

$$\text{PropInf2MemInf} : x_{\text{target}}, \mathcal{M}, \mathcal{D}_{\text{aux}}, \lambda \rightarrow \{member, non\text{-}member\} \tag{4}$$

where $\lambda$ is a set of $\mathcal{E}(\mathcal{M}(\mathcal{D}_{\text{aux}}))$ and the calibration function is $\lambda \frac{1}{N} \sum_i^N (\mathcal{P}_i - 0.5)$. Since PropInf in our scenario is a black-box attack, we can relax this information on both black/white-box MemInf. Specifically, different from PropInf2AttrInf, since the confidence of PropInf in this scenario is a constant number, there is no difference between empirical and theoretical settings.

## 4 THE ACE TOOLKIT

In this section, we present ACE, a modular toolkit designed to evaluate the above attack combos. Researchers have developed several software tools to measure the potential security/privacy risks of ML models, such as DEEPSEC (Ling et al., 2019) and CleverHans (Papernot et al., 2018) for evaluating adversarial example attacks, TROJANZOO (Pang et al., 2020) for backdoor attacks, as well as ML-Doctor (Liu et al., 2022b) for jointly analyzing the relationships among different attacks. Inspired by this work, we design a systematic framework to modularize our experiments better, namely ACE. To our knowledge, ACE is the first framework that jointly considers the combination of different inference-time attacks.

**Modules.** Fig. 2 illustrates the four modules of ACE:

1. **Input.** This module prepares the dataset and model for the other modules. More precisely, it performs dataset partition/preprocessing, constructs model architectures, and trains the model.

2. **Attack.** This module includes four inference-time attacks, each employing the most representative strategy. These attacks can be seamlessly replaced or updated with newer versions.

3. **Combo.** This module implements attack combinations where one support attack assists a primary attack. Currently, we have introduced four specific attack combination methods. Notably, users can add new combination methods as needed.

4. **Analysis.** This module evaluates and compares the performance of individual attacks and attack combinations. We include various evaluation metrics to provide a comprehensive analysis.

Overall, the modular design of ACE allows researchers and practitioners to reuse it as a standard benchmark tool, experimenting with new and additional datasets, model architectures, and attacks.

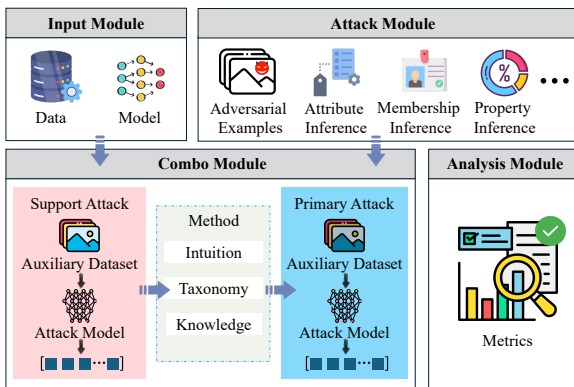

**Figure 2:** Overview of the workflow of ACE.

## 5 EXPERIMENTAL SETTINGS

We first select three benchmark datasets (see Section 5.1) and three state-of-the-art ML models (see Section 5.2) to train thousands of target and shadow models. For each dataset, we partition it into four parts (see Section 5.1), including the target training dataset, target testing dataset, shadow training dataset, and shadow testing dataset, to comply with the different scenarios discussed in Section 3.1.

### 5.1 DATASETS

In this work, we consider three benchmark datasets.

- **CelebA** (Liu et al., 2015) contains 202,599 face images, each labeled with 40 binary attributes. We select three attributes—*HighCheekbones*, *WearingNecktie*, and *ArchedEyebrows*—to define the target models' classes. The first two attributes form a 4-class classification for the first property, while the third attribute represents the second property.
- **CIFAR10** (Krizhevsky, 2009) is a widely used dataset containing 60,000 32x32 color images across ten classes, with 6,000 images per class. We group the second property into two categories: animal and non-animal.
- **Places** (Zhou et al., 2018) contains 1.8 million training images from 365 scene categories. The validation set has 50 images per category, and the test set has 900. For our study, we select 20 scenes, with 3,000 images each, and group them into two categories—indoor and outdoor—for the second property.

We divide each dataset into four parts. The first is the target training dataset. For PropInf, we randomly select samples based on the second property using different seeds to match the desired proportion. For other settings, we use the default proportion from the original dataset. The second part is the target test dataset, balanced across different properties. The third is the shadow training dataset, constructed similarly to the target training dataset. The fourth is the shadow test dataset, selected in the same way as the target test dataset. Note that this dataset splitting is the basic setup in this field (Shokri et al., 2017; Nasr et al., 2018; Salem et al., 2019; Liu et al., 2022b; He et al., 2022; Li et al., 2022; Liu et al., 2022a; Fu et al., 2023).

### 5.2 TARGET MODELS

We select three widely-used ML models, DenseNet121 (Huang et al., 2017), ResNet18 (He et al., 2016), and VGG19 (Simonyan & Zisserman, 2015). We set the mini-batch size to 256 and use cross-entropy as the loss function. We use Adam (Kingma & Ba, 2015) as the optimizer with a learning rate is 1e-2. Each target model is trained for 50 epochs. Note that for shadow models used in the MemInf and PropInf, we train thousands following the same process as the target models with the support of SecurityNet (Zhang et al., 2024).

**Table 1:** Performance of target models, namely, training/testing accuracy for each setting. We also provide the different proportions of the second property.

| Property Proportion | CelebA | | CIFAR10 | | Places | |
|---|---|---|---|---|---|---|
| | 2:8 | 5:5 | 2:8 | 5:5 | 2:8 | 5:5 |
| **DenseNet121** | 0.988/0.835 | 0.987/0.840 | 0.866/0.653 | 0.882/0.687 | 0.844/0.634 | 0.883/0.668 |
| **ResNet18** | 0.994/0.829 | 0.993/0.834 | 0.812/0.600 | 0.896/0.677 | 0.821/0.584 | 0.709/0.589 |
| **VGG19** | 0.935/0.833 | 0.937/0.845 | 0.764/0.565 | 0.843/0.645 | 0.842/0.668 | 0.878/0.677 |

## 5.3 Attack Models

**Attribute Inference.** At the preparatory level, the assistant from PropInf will not influence the types of inputs. Therefore, our attack model is a 2-layer MLP where its input is the embeddings from the second-to-last layer of the target model. We use cross-entropy as the loss function and Adam as the optimizer with a learning rate of 1e-2. The attack model is trained for 100 epochs. We use *accuracy* and *F1 score* for the evaluation metrics.

**Membership Inference.** Recall that there are four different scenarios for MemInf; we establish two types of attack models: one for the black-box and the other for the white-box setting. For black-box settings, our original attack model has two inputs: the target sample's ranked posteriors and a binary indicator on whether the target sample is predicted correctly. Each input is first fed into a different 2-layer MLP. Then, the two obtained embeddings are concatenated and fed into a 4-layer MLP. For the white-box, we have four inputs for this attack model, including the target sample's ranked posteriors, classification loss, gradients of the parameters of the target model's last layer, and one-hot encoding of its true label. Each input is fed into a different neural network, and the resulting embeddings are concatenated as input to a 4-layer MLP. We use ReLU as the activation function for the attack models. For the attack scenario assisted by ADV, the inputs of both the black-box and white-box attack models expand the $L_2$ distance between each image and its adversarial example in the auxiliary dataset. The original attack model remains the same for the attack scenario assisted by PropInf, but the encoder for choosing $\lambda$ is a 4-layer MLP. The attack model is trained for 50 epochs by using the Adam optimizer with a learning rate of 1e-5. We adopt *accuracy*, *F1 score*, *AUC score*, and *TPR @0.1% FPR* as the evaluation metrics.

**Property Inference.** Recall that the algorithm level needs to add additional information during the attack phase. For PropInf, the attack model is a meta-classifier; its inputs are organized from the unified overall outputs of each target (shadow) model by feeding the test auxiliary dataset with different proportions of another property. For the assisted PropInf, the inputs also expand a one-dimensional vector combo of the $L_2$ distance between each image and its adversarial example in the test auxiliary dataset. We adopt *accuracy* as the evaluation metric on 100 models.

## 6 Experimental Evaluation

### 6.1 Target Model Utility

First, we present target model utilities in Table 1. Based on previous work (Liu et al., 2022b), we define an overfitting level as the difference between its accuracy on the training and test datasets; the greater this difference, the more overfitting the model is. As shown, the overfitting levels in our target models are less than 0.250. On the other hand, we ensure a real-world scenario as much as possible to validate the effectiveness of our attack combo. Note that target models trained on datasets with a 2:8 proportion for the second property are used for PropInf, while a 5:5 proportion is used for other attacks.

### 6.2 Preparation Level

At this level, since we only need to change the data preprocessing phase, the subsequent training of the attack model will remain consistent with the original attack. In this case, our focus will be on preprocessing the dataset. As mentioned before, we demonstrate this attack level through PropInf2AttrInf.

**PropInf2AttrInf**. We first present the attack performance of PropInf2AttrInf by comparing it with the original AttrInf. Table 2 demonstrates the results of PropInf2AttrInf. We can find that the

**Table 2:** Performance of PropInf2AttrInf. Here, the empirical setting is based on the confidence (posterior) of PropInf, while the theoretical setting is the label of the prediction of PropInf.

| Model | Mode | CelebA | | CIFAR10 | | Places | |
|---|---|---|---|---|---|---|---|
| | | F1 Score | Accuracy | F1 Score | Accuracy | F1 Score | Accuracy |
| **DenseNet121** | Origin | 0.771 | 0.712 | 0.916 | 0.911 | 0.667 | 0.500 |
| | Empirical | 0.789 | 0.780 | 0.930 | 0.929 | 0.923 | 0.921 |
| | Theoretical | 0.782 | 0.783 | 0.930 | 0.930 | 0.916 | 0.914 |
| **ResNet18** | Origin | 0.779 | 0.736 | 0.667 | 0.500 | 0.667 | 0.500 |
| | Empirical | 0.790 | 0.772 | 0.895 | 0.894 | 0.901 | 0.895 |
| | Theoretical | 0.789 | 0.774 | 0.880 | 0.872 | 0.911 | 0.909 |
| **VGG19** | Origin | 0.742 | 0.664 | 0.911 | 0.905 | 0.915 | 0.910 |
| | Empirical | 0.757 | 0.747 | 0.918 | 0.921 | 0.937 | 0.937 |
| | Theoretical | 0.759 | 0.748 | 0.917 | 0.917 | 0.937 | 0.937 |

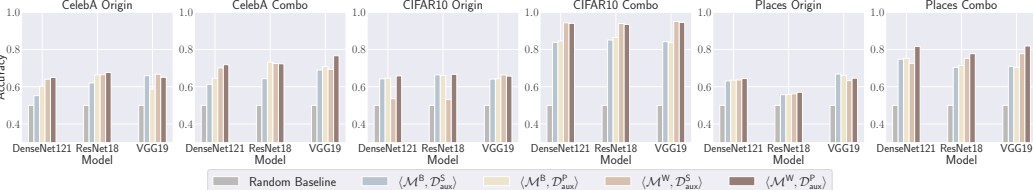

**Figure 3:** Accuracy of ADV2MemInf under different threat models, datasets, and target model architectures.

original AttrInf achieves a random guess for three scenarios. This indicates that simply collecting datasets will easily cause severe bias in property proportions, making original AttrInf challenging to achieve. Besides, the results are obviously better than the original attacks in both empirical and theoretical settings. For example, when using CIFAR10 to launch AttrInf on the DenseNet121 model, the original F1 score is 0.916, and accuracy is 0.911, while PropInf2AttrInf can achieve 0.930 and 0.929 for the empirical setting as well as 0.930 and 0.930 for the theoretical setting. This also means that with the assistance of PropInf, AttrInf can indeed achieve better results, which verifies our intuition: PropInf can better assist in determining the proportion of the target attribute in the original training dataset.

In addition, by training the PropInf attack model with 1,000 shadow models, the confidence of our target models exceeds 0.950. Therefore, there is little essential difference between our empirical and theoretical settings. In a nutshell, preprocessing in the preparatory phase is very intuitive, which requires us to choose a good assistant to complete.

### 6.3 EXECUTION LEVEL

At this level, we leverage ADV to assist two different types of attacks, MemInf and PropInf, during their execution stages.

**ADV2MemInf**. First, we evaluate the results of MemInf. We report the accuracy in Fig. 3 of ADV2MemInf, while Fig. 5, Fig. 6, and Table 5 in Appendix C report, respectively, F1, AUC score, and TPR @0.1% FPR. For some experiments, the original attacks do not achieve much higher attack performance than the random baseline, which means that overfitting does not have a significant impact on the attack (Shokri et al., 2017); see Appendix A.3. For instance, the original attack accuracy, F1 score, and AUC score of $\langle \text{MemInf}, \mathcal{M}^W, \mathcal{D}_{\text{aux}}^P \rangle$ on ResNet18 trained on Places are 0.544, 0.572, and 0.570, respectively. TPR @0.1% FPR score is 0.001, which is very low in this scenario. Compared to the previous works (Chen et al., 2020a; Leino & Fredrikson, 2020; Chen et al., 2021), white-box attacks have not significantly surpassed black-box attacks. This is expected because, in these works, the training accuracy of the target model can reach 1.000, meaning that for the training dataset, i.e., members, their loss is very close to zero. Nevertheless, this is not the case for non-members, allowing MemInf to achieve a high success rate. In contrast, since the training set accuracy does not reach 1.000 in our work, the loss may act as a form of noise in white-box attacks. We emphasize that our setting is more in line with real-world scenarios.

On the other hand, we find that ADV indeed significantly improves MemInf. For example, the combo attack accuracy, F1 score, and AUC score of $\langle \text{MemInf}, \mathcal{M}^W, \mathcal{D}_{\text{aux}}^P \rangle$ on ResNet18 trained on Places is 0.743, 0.653, 0.777, improved by nearly 0.200 compared to the original MemInf. TPR

**Table 3:** Performance of ADV2PropInf.

| Model | CelebA | | CIFAR10 | | Places | |
|---|---|---|---|---|---|---|
| | Origin | Combo | Origin | Combo | Origin | Combo |
| **DenseNet121** | 0.520 | 0.600 | 0.850 | 0.910 | 0.620 | 0.750 |
| **ResNet18** | 0.510 | 0.600 | 0.890 | 0.960 | 0.750 | 0.830 |
| **VGG19** | 0.540 | 0.630 | 0.860 | 0.930 | 0.730 | 0.750 |

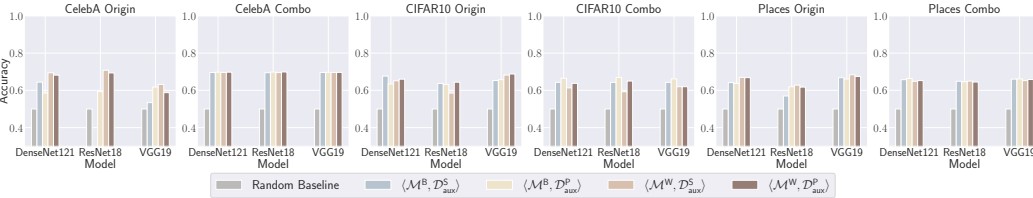

**Figure 4:** Accuracy of PropInf2MemInf under different threat models, datasets, and target model architectures.

@0.1% FPR score is also up to 0.490, indicating our combo attack model is effective at identifying true positives, even under very conservative conditions. More specifically, for the CelebA dataset, since we created a 4-class problem by combining the two labels of the first attribute, the ADV might not perform as well as on the other two datasets. This is because when noise affects one of the labels, it can change the combined class of the image, but this noise may not impact all the labels, leading to a smaller distance between members and non-members compared to the previous datasets. In general, the result first confirms our intuition; there is a distribution shift between the members and non-members when calculating the distance between the adversarial examples and the original data samples. In addition, for ADV, we believe that this distance has magnified the gap between members and non-members, resulting in an enhanced MemInf with a higher success rate. Therefore, the above results verify our intuition: there is a distribution shift between the members and non-members when calculating the distance between the adversarial examples and the original images.

**ADV2PropInf**. Next, we report our experimental results of ADV2PropInf in Table 3. We can clearly see that with the assistance of ADV, PropInf is significantly improved, which confirms our previous intuition. For example, the original PropInf on ResNet18 trained by CIFAR10 is 0.890 when using 100 shadow models. Nevertheless, after the assistance of ADV, the accuracy is increased to 0.960, equivalent to saving the time required to train at least 300 extra shadow models. Overall, the results of ADV2PropInf verify our intuition: for the auxiliary datasets with different proportions of the target property, the distribution of the $L_2$ distance between these samples and their adversarial samples should also be different.

## 6.4 EVALUATION LEVEL

At this stage, the support attack calibrates the results of the primary attack. In this work, we introduce PropInf to calibrate MemInf.

**PropInf2MemInf**. We report the accuracy of PropInf2MemInf in Fig. 4. In Appendix C, we also report F1 score and AUC score (Fig. 7 and Fig. 8) and, in Table 6, the TPR @0.1% FPR results. In many cases, the assistance of PropInf slightly improves MemInf's accuracy. While most TPR @0.1% FPR values remain near zero, there are instances where the combo attack achieves a higher TPR. For example, the combo attack on ResNet18 trained on CIFAR10 shows an accuracy of 0.669, F1 score of 0.731, and AUC of 0.656, compared to the original 0.631, 0.695, and 0.617. The TPR @0.1% FPR improves from 0.000 to 0.002. However, not all results show significant improvement. We attribute this to the general nature of the information from PropInf, which lacks the detailed insights that ADV provides for training the entire model. Without rich data or clear distinctions between members and non-members, improvements in metrics like F1 score and AUC are limited, suggesting that the original MemInf may already be near its upper bound. We attribute this to the general nature of the information from PropInf, which lacks the detailed insights that ADV provides for training the entire model. Improvements in metrics like F1 score and AUC are limited, suggesting that the original MemInf may already be near its upper bound. We also observe that with PropInf's support, attack performance remains stable across different scenarios (black-box and white-box), indicating that PropInf helps MemInf approach its performance limit. These results confirm our

intuition: a sample is more likely to be a member if it shares properties with most samples in the target group.

## 6.5 TAKEAWAYS

Overall, our evaluations demonstrate that combining different attack types significantly improves the effectiveness of primary attacks, leading to higher accuracy and success rates. These results confirm our earlier intuition about the benefits of attack combinations. Specifically, using ADV to assist MemInf and PropInf, as well as PropInf to assist AttrInf and MemInf, notably enhances the ability to identify training data and infer sensitive information.

## 7 RELATED WORK

More closely related to our work are studies focusing on the relationships between different types of attacks. Li & Zhang (2021) find a positive correlation between a sample's membership status and its robustness to adversarial noise. They leverage the differing adversarial noise magnitudes of members and non-members to mount a membership inference attack. However, our work significantly differs from theirs as we integrate one attack into another at different phases, using information from one attack to enhance or amplify another, while Li & Zhang (2021) relies on adversarial example information as the only signal for membership inference, without incorporating its original signal. Recently, Wen et al. (2024) proposed a method to strengthen membership inference through training-phase data poisoning attacks. However, data poisoning is a training-time attack, while membership inference occurs during the inference phase. We emphasize that although an attacker can launch attacks during both the training and inference phases, this assumption is prohibitively strong. As the first to systematically study the interactions between different attacks, we start only with the inference-time attack, as this is the most realistic scenario. Finally, Zhou et al. (2022) shows that property inference could enhance the performance of membership inference on GANs. However, their study focuses solely on GANs and proposes only one case study of attack combination. Furthermore, even though they provide valuable insight and inspire us to build ACE, their work lacks a high-level, systematic analysis of the intentional interactions among a more diverse set of attacks.

## 8 CONCLUSION

This paper provides the first step in exploring the intentional interaction between different types of attacks. Specifically, we focus on four extensively studied inference-time attacks: adversarial examples, attribute inference, membership inference, and property inference. To facilitate the study of their interactions, we establish a taxonomy based on three levels of the attack pipeline: preparation, execution, and evaluation, and propose four different attack combos: PropInf2AttrInf, ADV2MemInf, ADV2PropInf, and PropInf2MemInf. Extensive experiments across three model architectures and three benchmark datasets demonstrate the superior performance of the proposed attack combos.

Additionally, we introduce a reusable modular framework named ACE to integrate our attack combos. In this framework, we build four distinct modules to systematically examine the attack combinations. We believe that ACE will serve as a benchmark tool to facilitate future research on attack combos, enabling the seamless integration of new attacks, datasets, and models to further explore ML model vulnerabilities.

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

## A  INFERENCE-TIME ATTACKS

In this section, we present the four most representative attacks during the ML models' inference phase, namely, adversarial examples (Appendix A.1), attribute inference (Appendix A.2), membership inference (Appendix A.3), and property inference (Appendix A.4). Specifically, the first three are designed at the sample level, while the last one aims to infer the general information at the dataset level. Different attacks can be applied to different threat models; see Table 4. For each attack and each threat model, we concentrate on one representative state-of-the-art method.

**Table 4:** Different attacks under different threat models.

| Auxiliary Dataset | Model Access | |
|---|---|---|
| | Black-Box ($\mathcal{M}^{\mathsf{B}}$) | White-Box ($\mathcal{M}^{\mathsf{W}}$) |
| Partial ($\mathcal{D}_{\mathsf{aux}}^{\mathsf{P}}$) | MemInf | MemInf, AttrInf |
| Shadow ($\mathcal{D}_{\mathsf{aux}}^{\mathsf{S}}$) | MemInf, PropInf | MemInf, AttrInf |
| Query ($\mathcal{D}_{\mathsf{aux}}^{\mathsf{Q}}$) | PropInf | - |

### A.1  ADVERSARIAL EXAMPLES

Adversarial examples (ADV) (Szegedy et al., 2014; Goodfellow et al., 2015; Carlini & Wagner, 2017; Goodfellow et al., 2015; Papernot et al., 2016; Madry et al., 2018; Iyyer et al., 2018; Ribeiro et al., 2018; Alzantot et al., 2018; Belinkov & Bisk, 2018) are a type of ML security threat where malicious inputs are deliberately designed to deceive ML models. These inputs, known as adversarial examples, are typically crafted by making small, often imperceptible modifications to target data to cause the model to predict incorrectly. More formally, given a target data sample $x_{\mathsf{target}}$, (the access to) a target model $\mathcal{M}$, an adversarial example $x_{\mathsf{adv}}$ can be generated by applying a perturbation $\delta$ such that $x_{\mathsf{adv}} = x_{\mathsf{target}} + \delta$. To ensure it remains subtle, the perturbation is usually constrained by a norm $\|\delta\|_p \leq \epsilon$. The goal is to maximize the loss function $\ell\left(\mathcal{M}_\theta(x_{\mathsf{adv}}), y\right)$ In general, an adversarial attack can be defined as:

$$\mathsf{ADV} : x_{\mathsf{target}}, \mathcal{M} \rightarrow \{x_{\mathsf{adv}}\} \tag{5}$$

In general, this type of attack can be categorized into two types based on the knowledge of the adversary: black-box and white-box attacks ($\mathcal{M} \in \{\mathcal{M}^{\mathsf{B}}, \mathcal{M}^{\mathsf{W}}\}$).

**Black-Box** $\langle \mathsf{ADV}, \mathcal{M}^{\mathsf{B}}, x_{\mathsf{target}} \rangle$ (Andriushchenko et al., 2020). Black-box attacks operate under the assumption that the adversary has no internal knowledge of the models. Instead, the adversary can only observe the outputs from the model. This scenario is more common in the real world, where internal details are inaccessible. They usually leverage trial-and-error to approximate the gradient of the target model (Chen et al., 2020b) or randomized search schemes to approximate the boundary of the data samples (Andriushchenko et al., 2020).

**White-Box** $\langle \mathsf{ADV}, \mathcal{M}^{\mathsf{W}}, x_{\mathsf{target}} \rangle$ (Madry et al., 2018). White-box attacks assume the adversary has complete knowledge of the model, including its architecture, parameters, and training data. It

allows the adversary to precisely calculate the most effective perturbations to maximize errors of ML models, often employing gradient-based methods to manipulate the input data directly, such as C&W (Carlini & Wagner, 2017), FGSM (Goodfellow et al., 2015), JSMA (Papernot et al., 2016), and PGD (Madry et al., 2018).

## A.2 ATTRIBUTE INFERENCE

An ML model may inadvertently learn additional information during the training process unrelated to its original tasks. For instance, a model used to predict the ages from the profile photographs may also unwittingly acquire the capability to predict races (Melis et al., 2019; Song & Shmatikov, 2020; Liu et al., 2022b). Exploiting such unintended information leakage is known as attribute inference (AttrInf). State-of-the-art attacks usually rely on the embeddings of a target sample ($x_{\text{target}}$) obtained from the target model to predict the sample's target attributes. Thus, the adversary is assumed to have white-box access to the target model. Formally, attribute inference is defined as:

$$\text{AttrInf} : x_{\text{target}}, \mathcal{M}^{\text{W}}, \mathcal{D}^{\text{S}}_{\text{aux}} \rightarrow \{target\ attributes\} \tag{6}$$

where $\mathcal{D}_{\text{aux}}$ is an auxiliary dataset with the second attribute. The adversary is assumed to know the target attributes of the auxiliary dataset. They then use the target attribute embeddings of the auxiliary dataset to train the classifier to infer the actual dataset.

## A.3 MEMBERSHIP INFERENCE

Membership Inference attacks(MemInf) (Shokri et al., 2017) against ML models involve an adversary aiming to determine whether or not a target data sample is used to train a target ML model. More concretely, given a target data sample $x_{\text{target}}$, (the access to) a target model $\mathcal{M}$, and an auxiliary dataset $\mathcal{D}_{\text{aux}}$, a membership inference attack can be defined as:

$$\text{MemInf} : x_{\text{target}}, \mathcal{M}, \mathcal{D}_{\text{aux}} \rightarrow \{member, non\text{-}member\} \tag{7}$$

where $\mathcal{M} \in \{\mathcal{M}^{\text{B}}, \mathcal{M}^{\text{W}}\}$ and $\mathcal{D}_{\text{aux}} \in \{\mathcal{D}^{\text{P}}_{\text{aux}}, \mathcal{D}^{\text{S}}_{\text{aux}}\}$.

Membership inference has been extensively studied in literature (Shokri et al., 2017; Nasr et al., 2018; Salem et al., 2019; Jia et al., 2019; Sablayrolles et al., 2019; Li & Zhang, 2021; Chen et al., 2020a; Leino & Fredrikson, 2020; Chen et al., 2021; Liu et al., 2022b). Inferring membership of a target sample prompts severe privacy threats; for instance, if an ML model for drug dose prediction is trained using data from patients with a certain disease, then inclusion in the training dataset inherently leaks the individuals' health status. Overall, membership inference often signals that a target model is "leaky" and can be a gateway to additional attacks (Cervi, 2020).

In the following, we illustrate how to implement membership inference (MemInf) under different threat models.

**Black-Box/Shadow** $\langle \text{MemInf}, \mathcal{M}^{\text{B}}, \mathcal{D}^{\text{S}}_{\text{aux}} \rangle$ (Salem et al., 2019). We start with the most common and difficult setting for the attack (Shokri et al., 2017; Salem et al., 2019), whereby the adversary has black-box access ($\mathcal{M}^{\text{B}}$) to the target model and a shadow auxiliary dataset ($\mathcal{D}^{\text{S}}_{\text{aux}}$).

The adversary first splits the shadow dataset into two parts and uses one to train a shadow model on the same task. Next, the adversary uses the entire shadow dataset to query the shadow model. For each querying sample, the shadow model returns its posteriors and the predicted label: if the sample is part of the shadow model's training set, the adversary labels it as a member and, otherwise, as a non-member. With this labeled dataset, the adversary trains an attack model, which is a binary membership classifier. Finally, to determine whether a data sample is a member of the target model's training dataset, the sample is fed to the target model, and the posteriors and the predicted label (transformed to a binary indicator on whether the prediction is correct) are fed to the attack model.

**Black-Box/Partial** $\langle \text{MemInf}, \mathcal{M}^{\text{B}}, \mathcal{D}^{\text{P}}_{\text{aux}} \rangle$ (Salem et al., 2019). If the adversary has black-box access to the target model and a partial training dataset, the attack method is very similar to that for $\langle \text{MemInf}, \mathcal{M}^{\text{B}}, \mathcal{D}^{\text{S}}_{\text{aux}} \rangle$. However, the adversary does not need to train a shadow model; rather, they use the partial training dataset as the ground truth for membership and directly train their attack model.

**White-Box/Shadow** $\langle \text{MemInf}, \mathcal{M}^{\text{W}}, \mathcal{D}^{\text{S}}_{\text{aux}} \rangle$ (Nasr et al., 2019). Nasr et al. (Nasr et al., 2019) introduce an attack in the white-box setting with either a shadow or a partial training dataset as the

auxiliary dataset.[1] In the former, similar to $\langle \mathsf{MemInf}, \mathcal{M}^{\mathsf{B}}, \mathcal{D}^{\mathsf{S}}_{\mathsf{aux}} \rangle$, the adversary uses $\mathcal{D}^{\mathsf{S}}_{\mathsf{aux}}$ to train a shadow model to mimic the behavior of the target model and to generate data to train their attack model. As the adversary has white-box access to the target model, they can also exploit the target sample's gradients concerning the model parameters, embeddings from different intermediate layers, classification loss, and prediction posteriors (and label).

**White-Box/Partial** $\langle \mathsf{MemInf}, \mathcal{M}^{\mathsf{W}}, \mathcal{D}^{\mathsf{P}}_{\mathsf{aux}} \rangle$ (**Nasr et al., 2019**). The attack methodology here is almost identical to the black-box counterpart. The only difference is that the adversary can use the same set of features as the attack model for $\langle \mathsf{MemInf}, \mathcal{M}^{\mathsf{W}}, \mathcal{D}^{\mathsf{S}}_{\mathsf{aux}} \rangle$.

### A.4 PROPERTY INFERENCE

Property inference attacks ($\mathsf{PropInf}$) (Melis et al., 2019; Ganju et al., 2018; Mahloujifar et al., 2022; Zhou et al., 2022) aim to infer general information about the training dataset, such as the proportion of data with a specific property unrelated to the main classification task. For example, the gender ratio in the training dataset can be inferred when a model for classifying race is given. Previous works require access to the training process of the model (e.g., via gradients (Melis et al., 2019)) or to model parameters (Ganju et al., 2018). These methods are easy to implement for a few layers of neural networks. However, once the model becomes complex, the vast computational and memory resources are difficult to achieve. In addition, we build the query auxiliary datasets $\mathcal{D}^{Q}_{\mathsf{aux}}$ with different proportions of property. Therefore, in this paper, given a target model $\mathcal{M}$, the adversary first trains the shadow models by shadow auxiliary datasets $\mathcal{D}^{\mathsf{S}}_{\mathsf{aux}}$ with different proportions of the target property. Next, they query these shadow models to get the outputs of each proportion and concatenate these results together to train a meta-classifier for the property inference. We only need black-box access for this attack. Thus, the property inference can be defined as:

$$\mathsf{PropInf} : \mathcal{M}^{\mathsf{B}}, \mathcal{D}^{\mathsf{T}}_{\mathsf{aux}}, \mathcal{D}^{\mathsf{S}}_{\mathsf{aux}} \rightarrow \{target\ property\} \tag{8}$$

The global properties of a dataset are confidential when they relate to the proprietary information or intellectual property that the data contains, which its owner is not willing to share. This exposure can lead to severe privacy violations, especially if the data is protected by regulations like GDPR (European Union, 2016).

## B LIMITATION & DISCUSSION

**Limitations.** Naturally, our work is not without limitations. First, we focus on four inference-time attacks in the image domain. While attacks exist during the training phase, e.g., enhancing membership inference through backdoor attacks (Wen et al., 2024) or poisoning attacks (Chen et al., 2022), their settings are more complex, especially in real-world scenarios. More specifically, they typically require stronger adversarial assumptions, e.g., interfering with the training process or owning the training dataset.

We currently focus on image datasets because the types of attacks and their implementations are more detailed and comprehensive in image datasets. We also do not consider model stealing attacks, as they primarily, to some extent, convert black-box models to white-box models, which indeed can enhance the success rate of many attacks. Since we aim to explore the impact of attack combos during the attack's different phases, we emphasize that we do not change the overall attack process and the main attack approach.

**Potential Countermeasures.** A possible defense strategy against the attacks we consider is robust adversarial training, where models are trained on adversarial examples to improve robustness. Differential privacy techniques can also protect sensitive information by adding noise to the data, mitigating the risk of attribute, membership, and property inference attacks. Model ensembling, where predictions are aggregated from multiple models, can increase robustness by making it harder for adversaries to exploit vulnerabilities in a single model. However, we emphasize that, currently, no single defense can protect against all ML model attacks, and effective defenses against property inference or attribute inference are lacking (Liu et al., 2022b). As we focus on providing new in-

---

[1]The attack by Nasr et al. (2019) was originally designed for the partial training dataset setting, but it can be adapted to the shadow dataset setting.

sights and techniques for enhancing model security through attack combos, we leave the in-depth exploration of more effective defense mechanisms against them to future work.

## C  ADDITIONAL RESULTS

In this section, we report additional plots and tables to complement the analysis from the main body of the paper.

**Table 5:** TPR @0.1% FPR of ADV2MemInf.

| Model | Mode | CelebA | | CIFAR10 | | Places | |
|---|---|---|---|---|---|---|---|
| | | Origin | Combo | Origin | Combo | Origin | Combo |
| DenseNet121 | $\langle \mathcal{M}^B, \mathcal{D}_{aux}^S \rangle$ | 0.000 | 0.007 | 0.009 | 0.011 | 0.002 | 0.003 |
| | $\langle \mathcal{M}^B, \mathcal{D}_{aux}^P \rangle$ | 0.002 | 0.006 | 0.002 | 0.217 | 0.002 | 0.003 |
| | $\langle \mathcal{M}^W, \mathcal{D}_{aux}^S \rangle$ | 0.004 | 0.008 | 0.016 | 0.887 | 0.001 | 0.500 |
| | $\langle \mathcal{M}^W, \mathcal{D}_{aux}^P \rangle$ | 0.004 | 0.010 | 0.011 | 0.875 | 0.002 | 0.486 |
| ResNet18 | $\langle \mathcal{M}^B, \mathcal{D}_{aux}^S \rangle$ | 0.001 | 0.003 | 0.004 | 0.006 | 0.002 | 0.004 |
| | $\langle \mathcal{M}^B, \mathcal{D}_{aux}^P \rangle$ | 0.003 | 0.009 | 0.004 | 0.073 | 0.002 | 0.003 |
| | $\langle \mathcal{M}^W, \mathcal{D}_{aux}^S \rangle$ | 0.002 | 0.007 | 0.003 | 0.879 | 0.001 | 0.501 |
| | $\langle \mathcal{M}^W, \mathcal{D}_{aux}^P \rangle$ | 0.004 | 0.008 | 0.009 | 0.868 | 0.001 | 0.490 |
| VGG19 | $\langle \mathcal{M}^B, \mathcal{D}_{aux}^S \rangle$ | 0.001 | 0.006 | 0.002 | 0.074 | 0.002 | 0.004 |
| | $\langle \mathcal{M}^B, \mathcal{D}_{aux}^P \rangle$ | 0.001 | 0.011 | 0.003 | 0.239 | 0.002 | 0.009 |
| | $\langle \mathcal{M}^W, \mathcal{D}_{aux}^S \rangle$ | 0.001 | 0.008 | 0.016 | 0.902 | 0.001 | 0.500 |
| | $\langle \mathcal{M}^W, \mathcal{D}_{aux}^P \rangle$ | 0.001 | 0.009 | 0.002 | 0.899 | 0.001 | 0.494 |

**Table 6:** TPR @0.1% FPR of PropInf2MemInf.

| Model | Mode | CelebA | | CIFAR10 | | Places | |
|---|---|---|---|---|---|---|---|
| | | Origin | Combo | Origin | Combo | Origin | Combo |
| DenseNet121 | $\langle \mathcal{M}^B, \mathcal{D}_{aux}^S \rangle$ | 0.002 | 0.007 | 0.003 | 0.002 | 0.003 | 0.003 |
| | $\langle \mathcal{M}^B, \mathcal{D}_{aux}^P \rangle$ | 0.001 | 0.007 | 0.000 | 0.001 | 0.003 | 0.003 |
| | $\langle \mathcal{M}^W, \mathcal{D}_{aux}^S \rangle$ | 0.002 | 0.009 | 0.000 | 0.001 | 0.003 | 0.002 |
| | $\langle \mathcal{M}^W, \mathcal{D}_{aux}^P \rangle$ | 0.003 | 0.009 | 0.000 | 0.000 | 0.002 | 0.003 |
| ResNet18 | $\langle \mathcal{M}^B, \mathcal{D}_{aux}^S \rangle$ | 0.000 | 0.004 | 0.000 | 0.002 | 0.000 | 0.002 |
| | $\langle \mathcal{M}^B, \mathcal{D}_{aux}^P \rangle$ | 0.001 | 0.006 | 0.000 | 0.000 | 0.001 | 0.001 |
| | $\langle \mathcal{M}^W, \mathcal{D}_{aux}^S \rangle$ | 0.004 | 0.007 | 0.002 | 0.001 | 0.002 | 0.002 |
| | $\langle \mathcal{M}^W, \mathcal{D}_{aux}^P \rangle$ | 0.005 | 0.009 | 0.001 | 0.004 | 0.002 | 0.004 |
| VGG19 | $\langle \mathcal{M}^B, \mathcal{D}_{aux}^S \rangle$ | 0.001 | 0.010 | 0.001 | 0.001 | 0.001 | 0.004 |
| | $\langle \mathcal{M}^B, \mathcal{D}_{aux}^P \rangle$ | 0.001 | 0.014 | 0.000 | 0.003 | 0.002 | 0.001 |
| | $\langle \mathcal{M}^W, \mathcal{D}_{aux}^S \rangle$ | 0.001 | 0.013 | 0.001 | 0.002 | 0.001 | 0.001 |
| | $\langle \mathcal{M}^W, \mathcal{D}_{aux}^P \rangle$ | 0.001 | 0.015 | 0.005 | 0.000 | 0.004 | 0.001 |

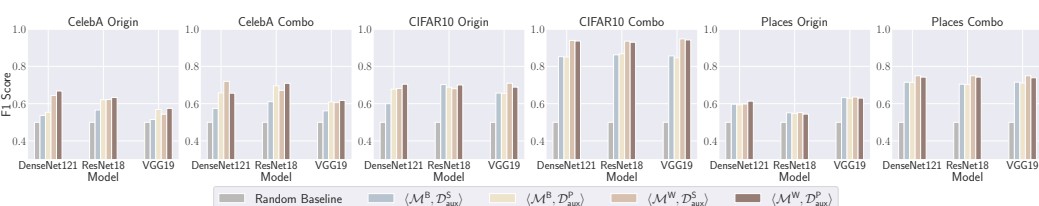

**Figure 5:** F1 score of ADV2MemInf under different threat models, datasets, and target model architectures.

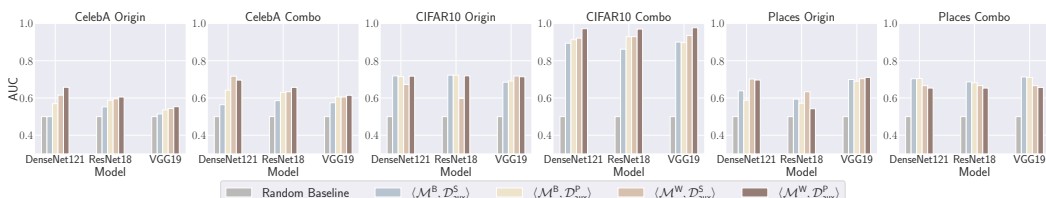

**Figure 6:** AUC of ADV2MemInf under different threat models, datasets, and target model architectures.

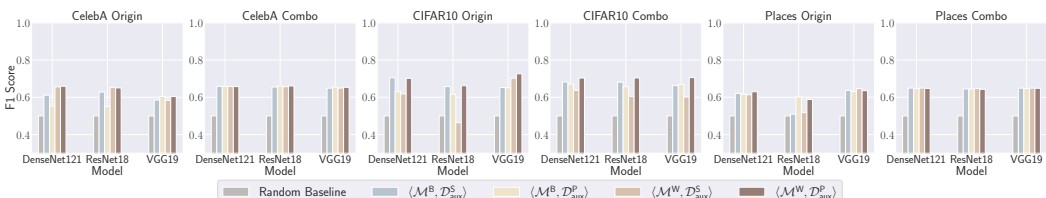

**Figure 7:** F1 score of PropInf2MemInf under different threat models, datasets, and target model architectures.

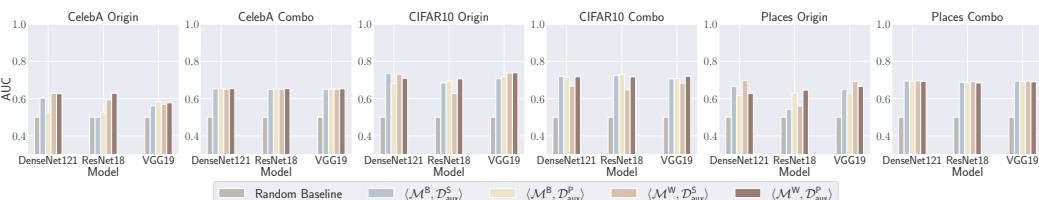

**Figure 8:** AUC of PropInf2MemInf under different threat models, datasets, and target model architectures.

