# OpenReview forum: "ACE: Attack Combo Enhancement Against Machine Learning Models"
_ICLR.cc/2025/Conference — ICLR 2025 Conference Withdrawn Submission_

### Official Review · Reviewer_MQJW · 2024-10-29

**Soundness:** 2
**Presentation:** 3
**Contribution:** 3
**Rating:** 5
**Confidence:** 5

**Summary:**

In this paper, the authors investigate intentional interactions among various inference-phase attacks against ML models and how an adversary can leverage the information obtained from one type of attack to increase the effectiveness of another attack. The proposed framework, ACE, combines various attacks in different phases of attack implementation (preparation, execution, and evaluation phases). Empirical results in three different datasets show that attack performance is increased when they are combined. I think ACE is a novel tool for sophisticated adversaries, but the paper lacks a systematic evaluation (see weaknesses and questions) and might need a revision to improve the quality/discussion.

**Strengths:**

1. ACE is a novel framework that combines different inference-time attacks with the aim to enhance each other's performance.
2. The rationale for each attack combination aimed at improving the primary attack is well-founded.
3. The experimental setup is clear enough to replicate experiments if necessary.

**Weaknesses:**

1. The paper focuses on improving the performance of the attacks using various attack combinations. However, the paper lacks an evaluation of how common defense mechanisms (against adversarial examples, membership inference, etc.) affect the effectiveness of such enhanced attacks.
2. Although the authors empirically show that the effectiveness of the attack increases when they leverage the information from another inference-phase attack, the paper lacks disussion on the efficiency. For example, the implementation of MemInf and PropInf requires the training of numerous shadow models. Although adversarial examples increase the effectiveness of MemInf, they do not reduce the attack's cost and further complicate the attack mechanism, potentially rendering it impractical as an attack strategy.
3. Although the authors consider both white-box and black-box settings, the population of attack strategies in the paper is not enough to establish a benchmark tool as detailed as, e.g., ML-Doctor or TrojanZoo.

**Questions:**

1. In ADV2MemInf, the authors chose to constrain the amount of perturbation using the $l_2$ norm. What is the reason behind this design choice, given that both Square and PGD are applicable with different $l_p$ norms? A broader question is: What kind of criterion are the authors using when choosing specific attacks during the experimental evaluation? For instance, using more recent adversarial example generation techniques than PGD might give better results when combined with MemInf.
2. Other prevalent inference-phase attacks are model extraction (or model stealing) and model inversion. Why have the authors chosen not to include these attacks within the ACE framework? For example, shadow models used in membership inference could potentially improve model extraction, or vice versa.
3. In Table 2, PropInf2AttInf in CelebA does not improve the F1 score (only 1 pp on average) and the VGG19 model trained on the CIFAR10 dataset. What is the reason behind this while the remaining results show some improvement?
4. In page 8, lines 419-420, the authors state that the overfitting does not have a significant impact on the membership inference attack. This conclusion is false; as demonstrated by other state-of-the-art works, overfitting, in fact, affects the membership inference attack (Shokri et al., 2017; Liu et al., 2022b).

---

### Official Review · Reviewer_o4G3 · 2024-11-01

**Soundness:** 3
**Presentation:** 3
**Contribution:** 3
**Rating:** 6
**Confidence:** 4

**Summary:**

This is the first study that investigates the combination of different types of test-time adversarial attacks, including adversarial examples, attribute inference, membership inference, and property inference. The authors decompose the attack pipeline into three phases, i.e., preparation, execution, and evaluation. Through experiments, the authors identify four effective combinations of attacks, such as property inference assisting attribute inference in the preparation phase and adversarial examples assisting property inference in the execution phase. The authors also develop a modular and reusable toolkit that helps investigate effective attack combinations.

**Strengths:**

- To the best of my knowledge, this is the first work to investigate attack combinations across different categories. This is important to develop more secure and reliable models.
- The released toolkit is beneficial to further exploring the threat of other kinds of attack combinations.
- The experimental results show that certain combinations of attacks perform better than existing attacks and are insightful for developing stronger attacks and more robust models.

**Weaknesses:**

- The architecture of the attacked model is limited. To support the generality of the findings, it would be helpful to attack Transformer-based models, which are known to behave differently than CNNs against adversarial attacks.
- The experiments are conducted on relatively small datasets. It would be helpful to investigate the behavior of attack combo on large-scale datasets.

**Questions:**

Please see the weaknesses above.

---

### Official Review · Reviewer_mw7x · 2024-11-03

**Soundness:** 2
**Presentation:** 2
**Contribution:** 2
**Rating:** 3
**Confidence:** 4

**Summary:**

This paper explores "attack combos" where multiple ML attacks are combined, enhancing overall impact. Traditional studies examine attacks individually, but adversaries often employ multiple strategies at once. Focusing on four attacks—adversarial examples, attribute inference, membership inference, and property inference—the authors introduce a taxonomy for attack interactions across three stages: preparation, execution, and evaluation. They identify four effective combos, demonstrating their effectiveness across various ML models and datasets. A toolkit, ACE, is also developed to support research in this area.

**Strengths:**

1. The attack vector is interesting and easy to understand
2. The attack setting is common in real-world applications
3. The results of the individual/combo attack seem promising

**Weaknesses:**

1. This paper violates plagiarism policies. According to iThenticate/Turnitin results, the authors directly copied the following content from a previous publication [1]: (1) Section 5.3, covering Attribute Inference and Membership Inference, and (2) Sections A.2 and A.3. The plagiarized material spans approximately one and a half pages, which is a substantial infringement of ethical guidelines.

2. The novelty of this paper appears limited. The authors primarily evaluate four standard attacks and their combinations on three basic datasets. Some key advanced attacks, such as model stealing and backdoor attacks, are equally critical in this domain and also need to be thoroughly evaluated and discussed.

3. The lack of critical details makes it difficult to understand what the authors did or what motivated their choices. For instance, while the paper claims that the combo attack is more effective than individual attacks, this is not clearly explained in the methodology and evaluation sections.

4. The paper does not clarify whether the proposed ACE tool can be used to evaluate the impact of inference attacks on commercial ML models, such as those provided through Machine Learning as a Service (MLaaS).

5. Writing needs to be improved. Please proofread the whole paper carefully to correct typos and grammar errors.

[1] Liu, Yugeng, Rui Wen, Xinlei He, Ahmed Salem, Zhikun Zhang, Michael Backes, Emiliano De Cristofaro, Mario Fritz, and Yang Zhang. "{ML-Doctor}: Holistic risk assessment of inference attacks against machine learning models." In 31st USENIX Security Symposium (USENIX Security 22), pp. 4525-4542. 2022.

**Questions:**

Please refer to my comments for more details.

**Details Of Ethics Concerns:**

This paper violates plagiarism policies. According to iThenticate/Turnitin results, the authors directly copied the following content from a previous publication [1]: (1) Section 5.3, covering Attribute Inference and Membership Inference, and (2) Sections A.2 and A.3. The plagiarized material spans approximately one and a half pages, which is a substantial infringement of ethical guidelines.

[1] Liu, Yugeng, Rui Wen, Xinlei He, Ahmed Salem, Zhikun Zhang, Michael Backes, Emiliano De Cristofaro, Mario Fritz, and Yang Zhang. "{ML-Doctor}: Holistic risk assessment of inference attacks against machine learning models." In 31st USENIX Security Symposium (USENIX Security 22), pp. 4525-4542. 2022.

---

### Official Review · Reviewer_B2VT · 2024-11-04

**Soundness:** 2
**Presentation:** 3
**Contribution:** 3
**Rating:** 6
**Confidence:** 4

**Summary:**

This paper studies the problem of conducting combined attacks by leveraging some secondary of different type to enhance the performance of the primary attack. For example, attackers can use property inference attacks first to better guess the target property distribution and then generate higher quality auxiliary datasets to further assist in the performance of the attribute inference attacks. The attack results indicate that the combined method outperforms the single attack, indicating the the strength of attacks in practice can be much more effective.

**Strengths:**

1. The idea of combining different types of attacks to enhance the performance of the primary attack is interesting.
2. The performed evaluations mostly cover the key points made in the paper.

**Weaknesses:**

1. The performance evaluation is heavily focused on leveraging shadow model based attacks. I think a comprehensive evaluation should also include some model-free attacks (e.g., LiRA for membership inference attacks), and this is important because, shadow model based approaches do not always outperform the model-free attacks in all settings. And the current combined method strongly binds to the shadow model based attack and so, the inclusion of the model-free attacks is necessary.
2. The evaluation of existing defense for the primary attacks should also be included. Some results such as showing that combined attacks can make the defense cost significantly higher (e.g., DP based defenses have to sacrifice more utility to empirically resist the attack effectiveness).
3. The related work section also misses some relevant works [1], [2], which both consider combining training and test-time types of attacks, given that the authors believe Wen et al.'s work as relevant.

[1] Feng & Tramer, "Privacy Backdoors: Stealing Data with Corrupted Pretrained Models", ICML 2024.

[2] Tian et al., "manipulating transfer learning for property inference", CVPR 2023.

**Questions:**

Adding additional evaluations with mode-free attack baselines as well as defense strategies will significantly improve the paper.

---

### Note · Authors · 2024-11-12

I have read and agree with the venue's withdrawal policy on behalf of myself and my co-authors.